# Improving Contraception Outreach through Human-Centered Design: A Pilot Study of the Ndingathe ("I Can") Intervention in Rural Malawi

Janelli Vallin[1]*, Martha Kamanga[2], Beth Phillips[1], Mandayachepa Nyando[3], Tamanda Jumbe[3], Innocencia Mtalimanja[3], Address Malata[3], Alfred Maluwa[3], Katherine Greenberg[1], Jenny X. Liu[1], Kelsey Holt[1]

1 University of California, San Francisco, California, United States of America, 2 Kamuzu University of Health Sciences, Lilongwe, Malawi, 3 Malawi University of Science and Technology, Limbe, Malawi

* vjanelli18@gmail.com

## Abstract

### Background

In Malawi, community health workers known as Health Surveillance Assistants (HSAs) can facilitate access to contraception through rural outreach. Self-injectable contraception also shows promise to facilitate contraceptive access, as women can store doses and re-inject on their own. We previously developed the *Ndingathe* ("I Can") intervention via human-centered design to strengthen contraception outreach by addressing HSAs' workflow challenges and enhancing self-injection (SI) counseling and support for interested clients. We piloted Ndingathe in two rural districts from June to December 2023.

### Methods

To assess the feasibility, acceptability, and potential effectiveness of Ndingathe, we conducted: pre- and post-surveys with 60 HSAs; 450 surveys with clients at HSAs' community outreach clinics; interviews with 40 clients, 20 HSAs, four health system stakeholders, and 20 experienced SI users who supported clients; and 20 observations of outreach clinics. We analyzed quantitative data using descriptive and inferential statistics. We conducted a thematic analysis of qualitative data.

### Results

Intervention components aimed at improving HSAs' workflow, including bicycles, lunch allowances, and workflow planning templates, were well-received by HSAs and health system stakeholders, and appeared to improve outreach clinic frequency and duration: the percentage of HSAs reporting conducting at least one outreach clinic per week rose from 65% to 95% after the pilot and observations suggested outreach

**Data availability statement:** All ICAN Malawi Program Evaluation dataset is available on the Harvard Dataverse: https://doi.org/10.7910/DVN/7JOZFI.

**Funding:** KH, JL Grant number: OPP1216593 Bill and Melinda Gates Foundation https://www.gatesfoundation.org/ The funders did not play any role in the study design, data collection and analysis, decision to publish, or preparation of the manuscript.

**Competing interests:** The authors have declared that no competing interests exist.

clinic hours extended into the afternoon, instead of ending before lunch, which clients appreciated. HSAs' average scores on the Role Conflict and Ambiguity in Complex Organizations and Role Overload Scales decreased after six months (2.9 to 2.3, $p < 0.0001$ and 3.7 to 3.0, $p < 0.001$, respectively). Both HSAs and clients positively received Ndingathe's SI mnemonic, designed to aid memory of SI steps, and support from experienced SI users during counseling. Clients' median self-reported fear of needles decreased from 3 (of 4) to 1 after interacting with an experienced user ($p < 0.001$). Clients felt reassurance when experienced SI users visited their homes for follow-up visits, conducted without HSAs, to offer support for SI. Challenges during the pilot included delays in lunch allowance disbursement, which impacted HSAs' morale and ability to expand outreach clinic hours.

## Conclusion

The Ndingathe intervention was feasible and acceptable from the viewpoint of multiple stakeholders in rural Malawi. Pilot findings suggest the intervention has the potential to improve contraception accessibility and reduce fear of self-injectable contraception.

## 1. Introduction

The ability to access contraception and decide what, if any, method to use is essential for individuals' reproductive rights [1]. Yet, access to contraceptives in rural areas of low- and middle-income countries (LMICs) remains limited due to persistent challenges, including inadequate healthcare infrastructure, [2] inconsistent supply chains [3], and sociocultural stigma related to contraceptive use stemming from religious and cultural beliefs and inequitable gender norms [4–6]. In Malawi, where over 80% of the population lives in rural areas [7], these barriers are particularly pronounced.

To improve the accessibility of contraception and other primary health care, Malawi has long relied on community health workers, known locally as Health Surveillance Assistants (HSAs), to extend services to underserved areas [8–10]. In rural areas far from primary care clinics, HSAs are often people's first point of contact for health-related issues and are instrumental in providing contraceptive services. However, studies have shown that HSAs often face systemic barriers limiting their ability to provide services, including limited transportation, inadequate financial support for proactive community outreach, and insufficient training or confidence in offering health counseling [11].

In addition to community health workers, self-care technologies hold promise for improving rural women's ability to use contraception when they want to. Research indicates that self-injectable depot medroxyprogesterone acetate (DMPA-SC) offers numerous benefits from the clients' perspective, including privacy and control [12]. Studies, such as those conducted in Ethiopia, Uganda, Malawi, and Pakistan, reveal high satisfaction among women who choose this method and choose to self-inject it [13–15].

Self-injection (SI) of DMPA-SC is also associated with higher contraceptive continuation rates than provider-administered DMPA-SC, suggesting SI may live up to its potential to allow women greater control over contraceptive use [16].

Despite the potential of DMPA-SC to ease contraceptive access barriers, the percentage of DMPA-SC users who self-inject in Malawi remains low (21% as of 2021), and qualitative research has identified fear of SI as a major concern among women [12,17]. Previous interventions focused on improving counseling to alleviate women's fears of SI in Malawi have shown promise but evidence is mixed on whether enhancing provider counseling is enough to encourage more women to try SI [18,19]. Approaches to simultaneously address service delivery challenges faced by HSAs serving rural areas and the psychosocial needs of clients to follow through on their interest in SI are lacking. Other research has high-lighted the need to not only ensure the availability of this contraceptive method option, but also to ensure women have support to overcome fears of injecting themselves and follow-up support for side effect concerns and to facilitate switching methods as needed [20,21].

To address this critical gap in evidence-based implementation models for improving contraceptive outreach clinic services in rural areas and providing social support for women interested in SI, we developed the Ndingathe ("I Can" in English) intervention through a participatory, human-centered design process in collaboration with community advisory boards [22]. Ndingathe is designed to work through two mechanisms: First, it aims to increase contraception service accessibility by addressing logistical barriers faced by HSAs that hinder their ability to conduct outreach clinics in the most rural communities. Second, it aims to enhance counseling and support for women interested in SI through a locally-developed counseling mnemonic to aid memory of SI procedures and by pairing HSAs with experienced SI users ("EUs") who support women during outreach clinics and afterwards in their homes upon request. By intentionally combining system-level supports for HSAs with peer support for women interested in SI, Ndingathe offers a novel, holistic approach to improving contraceptive access and strengthening women's ability to use SI if they prefer this method.

In this study, we sought to evaluate the feasibility, acceptability, and potential effectiveness of Ndingathe through a six-month pilot with an embedded mixed methods evaluation study in two districts in Malawi. Here, we report triangulated findings from surveys, interviews, and clinic observations with key participants and recipients of the pilot, including clients, HSAs, EUs, and other health system stakeholders.

## 2. Methods

### 2.1. Study design and setting

The pilot and evaluation took place in Ntchisi and Mulanje districts of Central and Southern Malawi, respectively. The Ndingathe intervention was piloted from June to December 2023, and follow-up study evaluation activities continued through February 2024. Ntchisi is predominantly rural, with the Chewa people being the largest ethnic group. Mulanje is more densely populated and ethnically diverse, with a blend of rural and semi-urban communities [23]. Specific communities in each district were selected, in conversation with local health officials, to maximize geographic spread of the intervention.

### 2.2. Description of the intervention

We report on the process of developing Ndingathe through a collaborative, human-centered design approach elsewhere [22]. Briefly, through research with HSAs and their clients in the two study districts, we identified specific needs and opportunities for improving contraception outreach clinic services and provision of DMPA-SC for SI in rural areas. Guided by these insights, we then held participatory design workshops, bringing together public health researchers, Malawian health officials, HSAs, and women with and without experience using SI from the community to brainstorm and refine ideas and develop and test prototypes of the ideas. We ultimately developed a streamlined solution, Ndingathe, that 1) optimizes the HSA workflow and 2) strengthens support for SI.

The HSA workflow component of Ndingathe consists of providing HSAs with a bicycle for transport and a lunch allowance to enable them to conduct outreach clinics in geographically isolated areas more frequently (at least once per week) and for the full working hours (8 am-4 pm) expected by the Ministry of Health (MOH) [24]. The lunch allowance given to HSAs as part of the intervention is intended to be used to purchase food at a local restaurant near the outreach clinic. Another workflow enhancement includes training HSAs to use a biweekly workflow planning tool with support from a supervisor. The tool encourages HSAs to proactively plan more community outreach clinics in rural areas. The strengthened SI support component consists of HSAs enlisting the assistance of two women living in the outreach clinic area with experience self-injecting DMPA-SC. These women (the "EUs") demonstrate the self-injection process and share their experiences with interested women during the outreach clinic. In the pilot, the EUs' role complemented the training provided by HSAs, who ultimately retained authority over validating women's correct SI technique during training. EUs also offer follow-up support to women who opt into a home visit soon after the outreach clinic and/or three months later for continued support related to self-injecting the next DMPA-SC dose. The SI component of the intervention also includes a locally-developed mnemonic for HSAs and EUs to teach women and commit to memory the critical steps in the SI process. The mnemonic is known as "Sakufima" in the local Chichewa language, which translates to "Shake, Close, Insert, Squeeze."

The Ndingathe intervention theory of action (Fig 1) depicts the two main pathways by which we posit that the intervention will improve women's contraceptive agency, defined as agency related to making and acting on contraceptive decisions [25], and increase use of self-injection. First, optimizing the HSA workflow is intended to improve service accessibility. Second, strengthening support for SI is intended to increase clients' self-efficacy and perceived control over contraceptive decision-making and self-efficacy to self-inject. It is important to note that Ndingathe does not aim to increase the percentage of women using SI, in line with rights-based principles for contraception care [26]. Rather, because DMPA-SC for SI was an under-utilized contraceptive technology when we developed Ndingathe, we hypothesized that enhanced SI service provision—important in its own right to ensure women are equipped with the knowledge and support for SI should they choose that option—would likely lead to increased use. The two-day Ndingathe training for HSAs and EUs covered work planning, principles of informed choice, and hands-on practice providing social support for women interested in SI. The training stressed the centrality of supporting women's contraceptive agency, regardless of

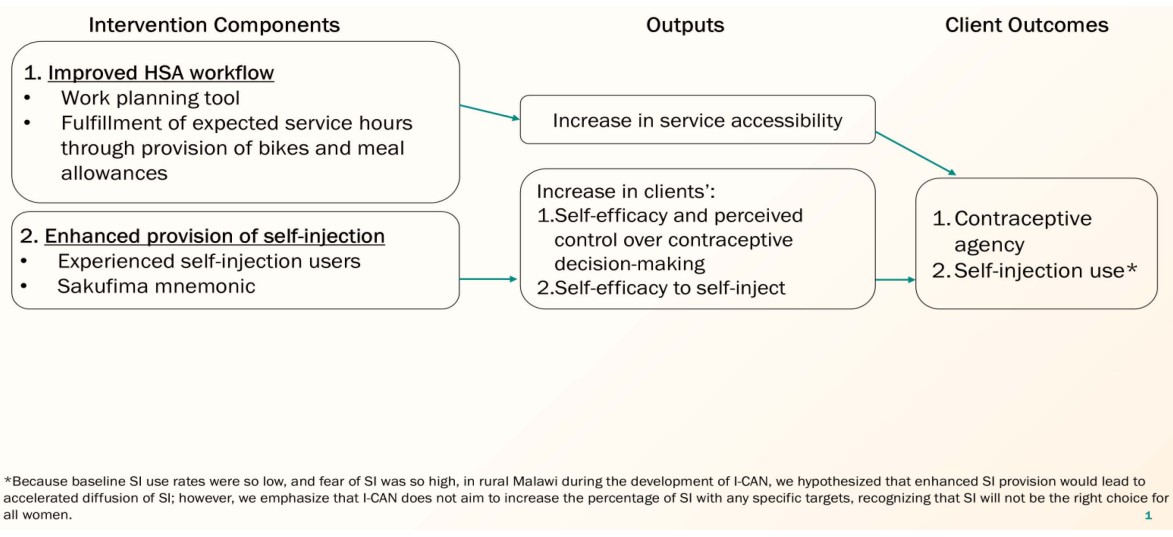

**Fig 1. Ndingathe intervention theory of action.**

what (if any) method a person selected during an outreach clinic, and equipped EUs to neutrally share their own experiences with SI without directing women to choose the method if not preferred.

## 2.3. Evaluation design

In total, 60 HSAs (30 from each district) were trained to deliver the Ndingathe intervention during the pilot period. HSAs were selected in consultation with the MOH based on the following criteria: 1) having previously received SI training and 2) living in the district. Half (n = 15 per district) of HSAs were paired with two EUs who alternated attending outreach clinics with the HSA (total EUs per district = 30). The other half of HSAs were not paired with EUs due to resource constraints, and a reduced intervention package was delivered without the EU, but including the improved workflow and SI mnemonic.

Alongside the pilot implementation, we collected mixed methods data from June 12th 2023 to February 28th 2024 to assess the intervention's feasibility, acceptability, and potential effectiveness from the perspective of multi-level stakeholders, including contraception clients, HSAs, EUs, and other health system stakeholders (Family Planning Coordinators and District Health Officers who play key roles in ensuring the delivery of contraception services in Malawi). The study was approved by the Malawi University of Science and Technology Ethics Review Committee (P.03/2020/007) and the University of California, San Francisco Institutional Review Board (23–38982). The ethics committees waived the need for parental consent for minors.

## 2.4. Sampling and data collection procedures

### 2.4.1. Surveys with outreach clinic clients.
Research assistants attended outreach clinics held by Ndingathe-trained HSAs to recruit clients for cross-sectional surveys during the pilot period. Research assistants selected a convenience sample of half of the trained HSAs in each district and attended 1–2 of their outreach clinics approximately halfway through the pilot to collect 10–20 client surveys per selected HSA. Among the 30 HSAs whose outreach clinics we targeted, half had EUs present. Research assistants approached all clients present on the day of data collection after they completed their visit with the HSA and screened for eligibility criteria: 1) having heard the HSA talk about SI that day and 2) not already using SI for contraception before the outreach clinic. For women interested in the survey, research assistants obtained written informed consent before administering the survey on tablets using Kobo Toolbox, an online survey platform. The survey was administered in a private outdoor location during the outreach clinic.

The survey collected data on client experiences receiving services from HSAs and EUs (where applicable) at the outreach clinic to assess acceptability. For clients who attended outreach clinics where EUs were present, the survey instrument included items assessing clients' self-reported fear of needles before and after they saw the EU using four-point Likert scales with responses ranging from "very afraid" to "very unafraid." These items were used to assess the intervention's potential effectiveness at improving women's self-efficacy to self-inject. The survey also included items asking how helpful the clients perceived the extended hours of the outreach clinic and the mnemonic to be (both asked on a four-point Likert scale ranging from "very unhelpful" to "very helpful"). The survey was administered by a research assistant in Chichewa and lasted 30–45 minutes. Each participant was given the equivalent of ~USD 8 to compensate for their time after they completed the survey.

### 2.4.2. Surveys with HSAs.
We conducted baseline and endline surveys with all 60 HSAs who participated in the pilot to assess their perception of the intervention and job-related experiences. On tablets, two trained research assistants collected HSAs' survey responses using Kobo Toolbox. HSAs were recruited to participate in baseline surveys at the beginning of the pilot training and in endline surveys during the closing ceremony where HSAs gathered to debrief at the end of the pilot. The surveys lasted approximately 15 minutes after RAs obtained written informed consent.

The HSA survey instrument included items assessing feasibility and acceptability of the intervention and measures of role conflict, role overload, and job satisfaction developed in the United States and previously adapted for use in Malawi [27]. Role conflict, when employees face contradictory expectations and struggle to meet demands, was measured

using the Role Conflict and Ambiguity in Complex Organizations Scale. This scale includes items measured on a five-point Likert scale (from "strongly disagree" to "strongly agree") in which higher scores denoted higher conflict; we used the 15-item version previously adapted for use in Malawi from the original 14-item scale [27]. Role overload, when an employee is tasked with more than they can handle, was measured with the Role Overload Scale [27]. This scale includes nine items measured on the same five-point scale, with higher scores denoting higher overload. Job satisfaction, when an employee is content with their job, was measured using the Minnesota Satisfaction Questionnaire. This scale includes 20 items measured on a five-point response scale ranging from "very dissatisfied" to "very satisfied," where higher scores denote more satisfaction [27]. The survey also asked HSAs to report the average number of contraception outreach clinics they conducted per week to assess the potential effectiveness of the intervention on improving service accessibility.

**2.4.3. In-depth interviews with clients, HSAs, experienced SI users, and other health system stakeholders.** To complement the quantitative survey data, we interviewed N = 40 women, 20 who had been surveyed about their experience at the outreach clinic (10 who interacted with an EU and 10 who only interacted with an HSA) and an additional 20 women who had received follow-up from an EU three months after their initial injection. We purposively sampled clients to ensure equal representation of women under the age of 25 and over the age of 25. We also conducted N = 20 interviews with HSAs who conducted the most number of outreaches (chosen to understand how the intervention enabled this increase), N = 20 interviews with a convenience sample of one of the two EUs they were paired with, and N = 4 interviews with each Family Planning Coordinator and District Health Officer from pilot districts. All interviews were conducted by a research assistant in Chichewa. Each participant provided written informed consent before the interview in their homes. The average length of the interviews was ~40 minutes. Participants received the equivalent of ~USD 8 in compensation for their time. Interviews were audio recorded, transcribed, and translated into English.

**2.4.4. Observations.** To assess the potential effectiveness of the intervention for encouraging HSAs to provide outreach clinics later into the afternoon, and to gauge the degree to which women attended later in the afternoon when given the option, we conducted 20 observations of outreach clinics throughout the pilot. Outreach clinics were selected for observation based on the workflow tools used during the intervention, aiming to capture a range of days, locations, and times to reflect typical service delivery across both districts. Research assistants were trained using a standardized observation protocol to ensure data collection and recording consistency. The research staff stayed at the outreach clinic the entire day and did not record identifiable information during the observations. A structured observation template guided data collection of the number of clients arriving in the morning versus the afternoon.

## 2.5. Data analysis

**2.5.1. Quantitative.** The HSA and client surveys were analyzed using Stata (Version 15). We generated descriptive statistics for all items. We used the Wilcoxon signed-rank test to evaluate the difference in clients' reports of their fear of needles before and after talking to an EU (among those who had interacted with an EU). We conducted paired t-tests to compare HSAs' role overload, role conflict, and job satisfaction scores pre- and post-pilot. We used McNemar's test to compare the percentage of HSAs reporting conducting at least one outreach clinic per week pre- and post-pilot. Given relatively small amounts of missing data, we performed complete case analyses. For observational data, we calculated descriptive statistics of the number of women observed attending outreach clinics in the morning versus the afternoon. No comparisons were made between groups due to limited resources.

**2.5.2. Qualitative.** The client interviews were analyzed using a line-by-line coding process facilitated by a codebook. We developed the initial coding framework deductively, grounded in the topics outlined in the interview guide related to acceptability and potential effectiveness of the intervention. A multidisciplinary team of five researchers (TJ, MN, MK, IM, JV) coded the transcripts using Dedoose, a qualitative data analysis software. This collaborative approach allowed us to incorporate diverse perspectives and ensure cultural relevance and sensitivity in the coding process. The team held initial

meetings to discuss coding discrepancies and refine the codebook. Following coding, the five researchers wrote up code summaries and developed themes.

We analyzed the stakeholder, HSA and EU interviews using a rapid framework analysis approach without formal coding. After transcription, two researchers (JV and KG) independently reviewed the interview transcripts to familiarize themselves with the data. Relevant excerpts were extracted and organized into a matrix around the pilot intervention's key components: the work planning tool, bicycles and lunch allowances, the mnemonic "Sakufima," and engagement with EUs. Within each element, excerpts were categorized according to feasibility and acceptability. Five researchers (IM, TJ, MN, JV, MK) were each assigned one component and used the matrix to develop themes. Organizing the data this way enabled systematic comparison across participant groups (e.g., clients, HSAs, stakeholders).

## 2.6. Inclusivity in global research

Additional information regarding the ethical, cultural, and scientific considerations specific to inclusivity in global research is included in the Supporting Information SX Checklist.

## 3. Results

### 3.1. Participant characteristics

More than half of the client survey sample was between the ages of 25–45 (51.7% in Mulanje and 60.8% in Ntchisi), while the rest of the sample was between the ages of 15–24 (48.3% in Mulanje and 39.2% in Ntchisi). More than half had a primary education (74.3% in Mulanje and 70.2% in Ntchisi) and 51.3% in Mulanje and 72.3% in Ntchisi attended the outreach clinic to switch methods; the client interview sample differed somewhat from the survey sample regarding age and education level (Table 1).

**Table 1. Characteristics of outreach clinic clients surveyed (N = 450) and interviewed (N = 40) during the Ndingathe pilot in Malawi in 2023.**

| Selected characteristics | Mulanje district | | Ntchisi district | |
|---|---|---|---|---|
| | Surveyed N (%) | Interviewed N (%) | Surveyed N (%) | Interviewed N (%) |
| **Age[1]** | | | | |
| 15-24 years | 109 (48.3) | 8 (20.0) | 88 (39.2) | 14 (70.0) |
| 25-45 years | 117 (51.7) | 12 (80.0) | 135 (60.8) | 6 (30.0) |
| **Education level[2]** | | | | |
| No formal education | 9 (3.9) | 0 (0.0) | 2 (1.0) | 0 (0.0) |
| Primary education | 168 (74.3) | 8 (40.0) | 156 (70.2) | 14 (70.0) |
| Secondary education | 49 (21.6) | 12 (60.0) | 64 (28.8) | 6 (30.0) |
| **Visit reason[3]** | | | | |
| Starting contraception for the first time | 38 (16.8) | Not collected | 28 (12.5) | Not collected |
| Continuing same method | 73 (32.3) | | 32 (14.2) | |
| Switching method | 116 (51.3) | | 162 (72.3) | |
| **Interaction at outreach clinic** | | | | |
| Only talked to an HSA | 105 (46.5) | 10 (50.0) | 104 (46.4) | 10 (50.0) |
| Additionally talked to an experienced user | 121 (53.5) | 10 (50.0) | 120 (53.6) | 10 (50.0) |

1 Age was missing for one survey participant.

2 Education was missing for two survey participants.

3 Visit reason was missing for one participant.

Most HSAs surveyed were male (70% in Mulanje, 60% in Ntchisi) with 10 or more years of work experience as an HSA (83.3% in Mulanje and 86.6% in Ntchisi) and all were between 34 and 54 years old; the interview sample was somewhat more female and less experienced than the survey sample (Table 2). We did not collect demographic information from the EUs or the Family Planning Coordinators and the District Health Officers interviewed from each study district.

The results of this evaluation are organized by intervention component. We first present themes related to the acceptability, feasibility, and potential effectiveness of the workflow components (work planning template, bicycles, and lunch allowance), followed by results on the additional SI support components (SI mnemonic and experienced SI users).

### 3.2. Evaluation of the workflow components of the Ndingathe intervention

**3.2.1. Potential effectiveness of workflow components.** Related to the first pathway in the intervention's theory of action (Fig 1) on workflow components increasing service accessibility, observational data during the pilot showed that HSAs stayed at outreach clinics into the afternoon, with 70% of women attending outreaches in the morning and 30% attending in the afternoon. No comparison was made to pre-pilot outreaches because HSAs did not previously work in the afternoons. Further, the percentage of HSAs who reported that they conducted an outreach clinic at least once a week increased from 65% at baseline to 95% after six months (Table 3).

Table 2. Characteristics of HSAs surveyed (N = 60) and interviewed (N = 20) during the Ndingathe pilot in Malawi in 2023.

| Selected Characteristics | Mulanje | | Ntchisi | |
|---|---|---|---|---|
| | Surveyed N (%) | Interviewed N (%) | Surveyed N (%) | Interviewed N (%) |
| **Gender** | | | | |
| Male | 21 (70.0) | 5 (50.0) | 18 (60.0) | 5 (50.0) |
| Female | 9 (30.0) | 5 (50.0) | 12 (40.0) | 5 (50.0) |
| **Age** | | | | |
| 34–44 years | 14 (46.6) | 6 (60.0) | 15 (50.0) | 3 (30.0) |
| 45–54 years | 16 (53.3) | 4 (40.0) | 15 (50.0) | 7 (70.0) |
| **Years of service** | | | | |
| 5–10 years | 5 (16.6) | 5 (50.0) | 4 (13.4) | 4 (40.0) |
| 10–20 years | 25 (83.3) | 5 (50.0) | 26 (86.6) | 6 (60.0) |

Table 3. Changes in HSA outcomes between baseline and six months (N = 60).

| Outcome | Baseline N (%) | Six months N (%) | p-value |
|---|---|---|---|
| Conducting an outreach clinic at least once per week | 39 (65.0) | 57 (95.0) | McNemar's Chi-square test p < 0.001 |
| | **Mean (SD)** | **Mean (SD)** | |
| Role Conflict and Ambiguity in Complex Organizations Scale Score (on a scale of 1–5)[1] | 2.9 (0.4) 95% CI = 2.8–3.0 | 2.3 (0.4) 95% CI = 2.2–2.5 | Paired t-test p < 0.0001 |
| Role Overload Scale Score (on a scale of 1–5)[2] | 3.7 (0.6) 95% CI = 3.5–3.8 | 3.0 (0.6) 95% CI = 2.9–3.2 | Paired t-test p < 0.001 |
| Minnesota Job Satisfaction Questionnaire Score (on a scale of 1–5)[3] | 4.1 (0.5) 95% CI = 3.9–4.2 | 4.0 (0.4) 95% CI = 3.9–4.1 | Paired t-test p = 0.72 |

1 Data missing for 6 participants at baseline and 1 participant at six months.

2 Data missing for 3 participants at baseline.

3 Data missing for 9 participants at baseline and 11 at endline.

**3.2.2. Acceptability of workflow components from the clients' perspective.** The lengthening of HSA outreach clinic hours into the afternoon was considered highly acceptable by clients. In surveys, 98% (N = 442) of clients found it helpful to have the option to attend an outreach clinic in the morning or the afternoon. HSAs announced outreach clinic hours at village meetings and posted flyers to indicate their working hours in places where the outreach clinic is normally conducted. In interviews, many clients described appreciating the extended outreach clinic hours because they could work or run errands in the morning and then access contraceptive services in the afternoon, providing greater convenience and flexibility in managing their daily responsibilities, as noted by this client:

*They did well because, in the past, when you miss [the outreach clinic] in the morning and get there by 11, you could find them [HSAs] gone. But now, it is good because you can go to the farm, or you can go to the market, and still you can find the health surveillance assistants right there. −22-year-old client, married, Ntchisi District*

**3.2.3. Acceptability and feasibility of workflow components from HSAs' and health system stakeholders' perspective.** We found that the work planning template was acceptable among HSAs, with many sharing in interviews that the introduction of structured work plans significantly enhanced the organization of their schedules. HSAs described that, before adopting these plans, they would plan their activities without a structured approach, leading to inconsistent service delivery of outreach clinics in the community. Before the intervention, most HSAs did not write down their schedule and would just visit the same place they normally provide outreach and not provide services elsewhere in their assigned catchment area. With a predefined schedule, they described planning and carrying out their duties more systematically throughout their assigned catchment area. Several HSAs from Mulanje described that the work plan was easy to follow:

*Having that [work planning template] directed us on where we were supposed to go every day. We worked so well, and the people were helped. – HSA, Mulanje District*

*It helped us to plan our programs in time and appropriately, and it was not difficult for us to follow because it was serving as a guide that we could easily know that this week, this is how we are going to work. -HSA, Mulanje District*

Several HSAs also noted that the workplan helped them proactively plan ahead for their work week; for example, one stated:

*The outreach was many because I could have a clinic for [injectables] per week and other clinics on later days, but I was not working under pressure. I had my fixed time such that the programs were not contradicting, because I was guided by the work plan. – HSA, Mulanje District*

HSAs expressed strong willingness to continue using the bi-weekly work plans even after the pilot, acknowledging its feasibility. Many HSAs mentioned that they would find a way to make or print their plans; for example, one stated:

*We realized that work planning is a good thing because the schedule will help us work easily. If the office does not supply, we have two or three copies which one can photocopy so that it can continue amongst us, instead of stopping altogether just because the pilot has ended.– HSA, Mulanje District*

Another HSA similarly responded when prompted if they would keep using the work planning model on their own without the printed version,

*Yes, it is possible to continue to use the bi-weekly plans even without printed weekly plan, we will improvise using the papers we have now. -HSA, Mulanje District*

From interviews with HSAs, bicycles and lunch allowances were also acceptable. Specifically, HSAs described that these facilitated their commute to outreach clinics and enabled them to stay longer at clinics without worries about transportation or sustenance. In Malawi, HSAs are faced with travelling long distances through challenging terrain, and unreliable availability of transport options. Several HSAs mentioned that these structural improvements afforded by the Ndingathe intervention allowed them to extend their outreach clinic services. For example, one noted:

*The lunch allowance helped us to visit sites far from our homes of residence. It helped us not to think of knocking off at 12 o'clock, to prepare lunch at home. We just go around the community, find food, eat, then get back to work until 4 o'clock. On the bike part, it helped us in transportation. From where we reside to the place of work, it helped us to reach our places of work in time, and when knocking off, reach our home of residence in time. – HSA, Ntchisi District*

However, some HSAs mentioned that delays in allowance payments from the study team created significant challenges, often leaving them to conduct multiple outreach clinics without the promised financial support. This disrupted pilot implementation and undermined morale and motivation, highlighting the importance of timely compensation to sustain effective service delivery. As expressed by one HSA:

*What I did not like was that we were told that we would be given the money at the beginning of the program, but sometimes we would come to the middle of the month without the allowance and maybe conduct two or three clinics without it, so it was disturbing.- HSA, Mulanje District*

HSAs interviewed highlighted that the bicycles significantly improved their commute, allowing them to reach distant villages more efficiently. After program completion, several HSAs mentioned that they would continue to use the bicycles to provide health services:

*We do outreach clinics and other tasks as health workers just because of the bike. And if the village is located at a long distance, the bike would help us. We can say that while the program has ended, the bike is still there and we are still using it for our purpose – HSA, Ntchisi District*

This continued use of bicycles underscores their critical role in supporting ongoing health outreach efforts, demonstrating the feasibility and sustained impact of integrating bicycles into health outreach initiatives. In response to questions about the sustainability of the intervention components, health system stakeholders and HSAs mentioned that in order to effectively carry out their work, there needs to be ongoing support from the District Health Management Team and sufficient resources, such as stationery, bicycles, and lunch allowances.

*We need stationery, photocopiers, and printers. Only one photocopier is working at this whole facility, making it difficult to provide work plans. – HSA, Mulanje District*

Related to HSAs' perception of their job, we found differences in role overload and role conflict between baseline and endline. Specifically, there was a significant decrease in the Role Conflict and Ambiguity in Complex Organizations Scale scores, from 2.90 to 2.30 ($p < 0.0001$) (Table 3). We also found a significant decrease in scores on the Role Overload Scale, from 3.7 to 3.0 ($p < 0.001$); scores on the Minnesota Job Satisfaction Questionnaire remained stable (from 4.1 to 4.0; $p = 0.72$).

### 3.3. Evaluation of the strengthened SI support components of the Ndingathe intervention

**3.3.1. Potential effectiveness of experienced SI users.** Related to the second pathway depicted in the intervention's theory of action (Fig 1), we found that EUs may improve women's self-efficacy to self-inject. In cross-sectional surveys

with clients, the median reported fear of needles was 3 (on a scale of 1–4) before talking to an EU, but dropped to 1 after talking to an EU (p < 0.001) (Table 4).

In interviews, many clients described initial anxiety or fear about self-injecting, particularly related to uncertainty about performing the technique correctly or concerns about potential side effects. However, they described that the presence of EUs during outreach clinics played a significant role in reducing this fear. Clients consistently emphasized that EUs provided emotional reassurance, normalized the self-injection experience, and built trust through their personal stories and peer modeling. For example, one client noted:

*We believed their words that we would be okay. They removed fears from us. – 25-year-old client, single, Ntchisi District*

Another client shared how seeing the EUs' confidence in their own bodies helped shift her own perceptions:

*[The EUs] explained their experiences using [SI]. They didn't face any issues with it; all showed they were fine in their bodies. Talking to them was encouraging and useful for us. – 31-year-old client, married, Ntchisi District*

This peer-based reassurance was not only meaningful to clients but also recognized by HSAs as a powerful complement to provider-led instruction. Several HSAs reported that EUs were able to reassure women more effectively than they could on their own, and potentially also made their workload more efficient. One HSA from Ntchisi reflected:

*What made the work easier was that after we taught them, they were also meeting up with the experienced users. Unlike when I was alone, it was taking time for a woman to self-inject. But when she sees the experienced user, they were motivated that this one is my fellow woman, then self-inject. – HSA, Ntchisi District*

By seeing women "like them" successfully self-injecting, clients felt more confident in their own abilities. This theme of peer reassurance and shared experience emerged frequently across interviews, underscoring the relational nature of learning and the unique value of integrating EUs into community-based SI programs.

**3.3.2. Acceptability of the SI mnemonic and experienced SI users from the perspective of clients.** The mnemonic was widely accepted by clients, with 95% (n = 429) reporting in surveys that it helped guide them through what the self-injection process would be like. Moreover, the mnemonic's localization in the local language enhanced its clarity and relatability, fostering stronger comprehension among clients. One client articulated how the mnemonic's components translated directly into practical steps, making the technique more approachable:

*It is helpful because SA [from the mnemonic "Sakufima"] means mixing the medicine, press slowly, in that way it means we are pressing the injection until it finishes, so we saw that these words are good. – 37-year-old client, married, Mulanje District*

**Table 4. Self-reported fear of needles before and after talking with an experienced SI user, cross-sectional survey data collected from outreach clinic clients during pilot (N = 240[1]).**

| Construct | Before talking to an experienced SI user Median (Interquartile range) | After talking to an experienced SI user Median (Interquartile range) | P-value |
|---|---|---|---|
| Reported fear of needles (on a scale of 1–4) | 3 (2) | 1 (0) | Wilcoxon Signed Rank Test p-value<0.001 |

1 Data missing for one participant who interacted with an experienced SI user during the pilot.

This culturally adapted mnemonic thus served not just as a memory aid but also as a source of reassurance, translating technical steps into familiar language and concepts. Alongside the mnemonic, interaction with EUs emerged as a critical source of encouragement and confidence for clients, many of whom initially doubted their ability to self-inject correctly. The combination of step-by-step guidance and real-time reminders from EUs, often reinforcing mnemonic cues, were described as invaluable. Clients recognized that this peer support helped bridge knowledge gaps and mitigate anxieties:

*It was helpful that the experienced user was reminding us to shake [the injection device] vigorously. Then they said 'handle it properly' as we injected ourselves. In simple terms, they were reminding us of the process. To me, I think there was a benefit because without the experienced user we would not know. – 28-year-old client, married, Ntchisi district*

Regarding follow-up support, clients who agreed to, and received follow-up visits at the time of the next dose (3 months) from EUs without the presence of an HSA, expressed appreciation for these visits by the EUs. They described these visits as providing an additional layer of support, reassurance, and continuity of care, making clients feel valued and cared for in their SI journey, as expressed by these two clients:

*I felt happy and remembered because the EU followed up with me. If it were others, they might have remained silent after the self-injection. – 28-year-old client, married, Ntchisi District*

*The expert woman visited us at home, asked about any problems, and ensured our well-being, which was very reassuring. – 22-year-old client, married, Ntchisi District*

Clients appreciated the reminders of the self-injection procedure and additional guidance from EUs, which helped them maintain proper self-injection practices. Clients in Ntchisi and Mulanje both noted in the interviews the helpfulness of the reminders that they received:

*The EUs helped us a lot by reminding us of the steps during reinjection, making us feel supported. – 32-year-old client, married, Mulanje District*

*I was very happy because when they came it was like they have reminded me what we used to learn when we meet them, so, they reminded me once again and I was refreshed. It was helpful because sometimes a person forgets due to being busy, so they reminded me that I have remembered what I am supposed to do and follow it. – 22-year-old client, married, Ntchisi District*

Finally, the interpersonal aspect of EU follow-up fostered trusting connections, making clients feel comfortable reaching out for support and making the process of self-injecting seem less intimidating. Several clients contrasted the ease of communication with EUs against the more formal and sometimes cumbersome hospital environment, as described by these clients:

*I was happy to open up to these women. If it [the services she was receiving] were at the hospital, it would have been a long process, but I could just call them when needed. – 36-year-old client, married, Mulanje District*

*I felt good, and my husband also felt honored to be present during the visit. It made us feel supported. – 38-year-old client, married, Mulanje District*

This relational trust not only improved adherence, but also enhanced the overall acceptability of the self-injection program by creating a supportive community network. As one experienced user notes:

*After three months, we followed them for encouragement and reminded them of their next date to self-inject. Some women had already self-injected while some said it was good that we reminded them [about their next self-injection date]. – 30-year-old experienced user, married, Ntchisi District*

### 3.3.3. Acceptability and feasibility of the mnemonic and EUs, from HSAs' and EUs' perspective.

Related to the mnemonic, we identified two key themes. First, HSAs reported that women generally reacted positively to the acronym, finding it interesting and enjoyable and facilitating their engagement with the self-injection process. In terms of cultural resonance and memorability, HSAs reported the Sakufima mnemonic was easy for women to recall because it sounded like a person's name or song, making it familiar and appealing.

The second theme was training efficiency. HSAs frequently mentioned that using the acronym saved time. It reduced the need for lengthy explanations and repetitive training sessions, as women could quickly grasp and remember the steps, as stated by this HSA:

*It was timesaving because, instead of explaining the steps, it was in short form, and people understood easily. When teaching a woman this mnemonic, she was paying full attention to understand, so we were saving time because few questions were asked, showing that she had understood at the beginning. – HSA, Mulanje District*

One HSA, interviewed after pilot completion, emphasized the ongoing use of the mnemonic, stating,

*We are continuing in spite of the program ending…. we are telling women about Sakufima so that they should not have difficulties remembering how to self-inject. – HSA, Mulanje District*

Regarding the EU component, not only were they perceived to be feasible by HSAs, but many HSAs reported that their work was easier and more effective when partnered with EUs at the outreach clinics and that EUs helped create a supportive environment in which clients felt more comfortable and open to learning about and trying self-injection. EUs themselves reported playing a powerful role in fostering both personal and community-level change. Many described feeling empowered as role models, taking pride in helping other women gain confidence in self-injection. Many EUs also noted that by sharing their own experiences, clients became more open to sharing their fears. In some cases, women were more open and forthright with the EUs compared to HSAs. As one EU described:

*What made me happy is that I was brave enough to self-inject, which made other women motivated to do the same. –29-year-old experienced user, married, Ntchisi District*

Because EUs lived in the same villages as clients, their support was easily accessible and embedded in community life, making follow-up feel informal yet reliable. Finally, their strong desire to continue supporting others even after the pilot ended reflects a sense of community ownership and promise of sustainability. EUs, clients and HSAs also continued to use the mnemonic, Sakufima, beyond the intervention period, as reported spontaneously during follow-up interviews (not tracked systematically).

## 4. Discussion

Our mixed methods evaluation of a pilot of the Ndingathe intervention found Ndingathe to be feasible to implement and acceptable from the point of view of women, HSAs, and EUs of SI in two rural study sites in Malawi. Our findings also suggest that the intervention may be effective in improving the accessibility of outreach services and in reducing women's fear of self-injection, both of which, per the intervention's theory of action, can increase women's agency related to contraceptive decisions and help diffuse self-injectable DMPA-SC as a novel self-care technology. The community-engaged human

centered design process used to develop Ndingathe is a strength that likely contributed to this pilot's strong acceptability findings, as locally appropriate components were prioritized in the design phase [22].

Ndingathe is unique among interventions to improve contraceptive care by addressing fundamental structural issues that restrict women's access to community health workers in the most rural areas [5]. The statistically significant decreases in HSAs' role conflict and role overload after participating in the pilot represent meaningful changes important not only for HSAs' quality of life but also signal potential to improve HSAs' ability to offer quality healthcare to clients in rural areas. Indeed, we observed an increase from just under two-thirds to almost all HSAs reporting providing contraception outreach at least once per week—a large shift that could substantially bolster contraceptive access in the most remote areas if sustained. Pilot findings suggest Ndingathe helped mitigate transportation challenges by equipping HSAs with bicycles, which allowed them to reach remote areas more frequently and stay for longer when also facilitated with lunch allowances. Bicycles have shown promise for improving community health outreach in other rural settings in sub-Saharan Africa [28]. Furthermore, providing HSAs with lunch allowances was appreciated by HSAs in the pilot, and has been demonstrated in other studies to improve the efficiency of community health workers by addressing the financial and physiological barriers they face when conducting outreach [29]. Our finding that the delayed lunch allowances affected HSAs' morale demonstrates that paying attention to workers' basic needs enables them to properly carry out their job duties [30].

Ndingathe also showed promise for diffusing contraceptive SI technology, as evidenced by a substantially lower client-reported fear of injecting oneself after talking to an EU compared to before. Hands-on support from experienced peers innovates on existing SI programs that focus primarily on spreading awareness of the option to self-inject and improving provider counseling [17,18]. A focus on peer support aligns with systematic reviews documenting that peer support may enhance the acceptability of new health behaviors by creating a trusting and relatable environment, especially for rural or other groups not well reached by the healthcare system [31,32]. In line with other research emphasizing the need for approaches to provide women continued support and follow-up for SI, particularly in settings where access to formal healthcare is limited [19], these pilot results suggest the follow-up visits conducted by EUs as part of Ndingathe may help women initiate or continue SI when they want to. Further, continued dissemination of the SI mnemonic by HSAs after the pilot, underscores the mnemonic's perceived value, ease of integration into routine practice, and suggests promise for long-term feasibility in contraception programs.

The need to enhance contraceptive outreach clinics provided by HSAs reflects broader trends in task-shifting in global health, where community health workers take on expanded responsibilities to alleviate workforce shortages and reduce inequities in contraceptive access between rural and urban populations [8]. However, just as they are tasked to assist with the healthcare shortages in underserved areas, HSAs also face challenges that circumvent their ability to provide these much-needed services, including lack of transport while working in remote regions, limited supportive supervision, and frequent stockout of medical supplies [8–10]. In our study, the addition of EUs working alongside HSAs eased the burden on HSAs, allowing them to work more efficiently and serve the needs of their clients, most of whom were from the same community as the EUs. This peer-driven approach helped to alleviate the time pressures faced by HSAs, echoing findings by Ballard et al. that collaboration between community health workers and peer supporters may enhance the effectiveness of healthcare delivery [33].

While the Ndingathe pilot showed promising results, several limitations should be considered. Most notably, the study lacked a control group. This major constraint limits our ability to attribute observed changes directly to the intervention, as we cannot rule out confounding. Further, our quantitative analyses did not account for potential clustering of survey responses within a district (or, in the case of client surveys, by HSA). As such, our findings should be considered exploratory and would benefit from further validation using experimental or quasi-experimental methods. Second, we did not measure client self-efficacy and perceived control over contraceptive decision-making—an output from the theory of action directly related to the hypothesized impact on contraceptive agency [25], which merits further investigation. Third,

the development and piloting of Ndingathe was done in two specific rural districts in Malawi, and generalizability to other areas of the country or other rural settings in sub-Saharan Africa is unknown. Fourth, stockouts of DMPA-SC occurred for about a month during the intervention period, making it difficult to offer interventional support for SI as intended. HSAs took it upon themselves to share extra stock amongst each other, allowing us to present our findings despite the interruption in stock. Fifth, because we prioritized interviewing HSAs who conducted the most outreaches to understand how Ndingathe may have facilitated this, we were limited in our ability to describe the factors that contributed to some HSAs conducting fewer outreaches. Sixth, we used convenience sampling for the client surveys, which may have introduced selection bias if participants did not represent all clients who attended clinics across the pilot period. Lastly, we reimbursed clients, EUs, and stakeholders for their time answering interview questions, which may have influenced their response and introduced courtesy bias.

## 5. Conclusion

These findings from a pilot of the Ndingathe ("I Can") intervention in Malawi show promise that Ndingathe could help address critical gaps in contraceptive access and support for women and adolescents in rural Malawi. This intervention contributes to filling a gap in approaches that simultaneously address community health workers' structural challenges in the most rural areas and enable greater psychosocial support for clients via a locally-derived mnemonic for self-injection and offering peer counselors who can help women feel confident to choose and perform self-injection if preferred. While these initial results are encouraging, further testing is warranted to establish the solution's effectiveness in achieving its intended outcomes, such as improved contraceptive agency, and to evaluate the minimum amount of resources required to ensure sustainability of the model. Future work should also explore adaptations of the Ndingathe intervention for other settings facing similar contraceptive access challenges.

## Acknowledgments

We acknowledge the district health management teams and the Ministry of Health in Malawi, in particular Jessie Chirwa, for their support in engaging with the Ndingathe pilot. We would like to thank Emas Potolani, Josephine Changole, Rabecca Bika, and Chifundo Zimba for their role as supervisors of the Ndingathe pilot.

## Author contributions

**Conceptualization:** Address Malata, Jenny X. Liu, Kelsey Holt.

**Data curation:** Janelli Vallin, Martha Kamanga, Mandayachepa Nyando, Tamanda Jumbe.

**Formal analysis:** Janelli Vallin, Martha Kamanga, Mandayachepa Nyando, Tamanda Jumbe, Innocencia Mtalimanja, Katherine Greenberg.

**Funding acquisition:** Address Malata, Jenny X. Liu, Kelsey Holt.

**Investigation:** Tamanda Jumbe, Innocencia Mtalimanja.

**Methodology:** Janelli Vallin, Jenny X. Liu.

**Project administration:** Janelli Vallin, Martha Kamanga, Mandayachepa Nyando, Tamanda Jumbe, Alfred Maluwa.

**Supervision:** Janelli Vallin, Martha Kamanga, Beth Phillips, Mandayachepa Nyando.

**Writing – original draft:** Janelli Vallin, Kelsey Holt.

**Writing – review & editing:** Janelli Vallin, Martha Kamanga, Beth Phillips, Mandayachepa Nyando, Tamanda Jumbe, Innocencia Mtalimanja, Address Malata, Alfred Maluwa, Jenny X. Liu, Kelsey Holt.

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
