## [Decision Letter · Decision Letter 0]

12 May 2025

PONE-D-25-06248Feasibility, acceptability, and potential effectiveness of a human-centered design-derived intervention to improve community health workers’ contraception outreach in rural MalawiPLOS ONE

Dear Dr. Vallin,

Thank you for submitting your manuscript to PLOS ONE. After careful consideration, we feel that it has merit but does not fully meet PLOS ONE’s publication criteria as it currently stands. Therefore, we invite you to submit a revised version of the manuscript that addresses the points raised during the review process.

This study is novel and important, with great potential to improve accessibility and use of contraceptives among hard-to-reach populations of women in Malawi. The project can also be easily translated to other sub-Saharan African countries. However, the manuscript needs strengthening before it can be accepted for publication. Please address the reviewers’ comments and consider the following points in your revised manuscript:

**Abstract**: Add a sentence or two about data analysis.**Introduction, **Page 4, Lines 67-69: The current text states, "These workers are often people’s first point of contact for health-related issues and are instrumental in promoting and providing contraceptive services, among many other primary care services." This should be reworded to clarify that Health Surveillance Assistants (HSAs) are the first point of contact for health-related issues in areas far from health centres (Primary health care delivery settings). This distinction is important because medical assistants are typically the first point of contact for health-related problems in rural areas. The primary role of HSAs is to promote health and prevent illnesses.**Methods (Description of the intervention)- **correct the typo on page 7, line 136, where 'Ndingathe' has been written as 'Ngindagthe'. Also, could you please provide more information about the contents of the observation template? Was the template structured or unstructured?**Data Analysis**: Please provide some details on how the framework analysis was used to analyse the transcripts and develop the themes. Could you also explain how data from observation were analysed?

We look forward to receiving your revised manuscript.

Kind regards,

Fatch Welcome Kalembo, Ph.D

Academic Editor

PLOS ONE

Journal Requirements:

3. We note that your Data Availability Statement is currently as follows: All relevant data are within the manuscript and in Supporting Information files.

4. Please remove your figures from within your manuscript file, leaving only the individual TIFF/EPS image files, uploaded separately. These will be automatically included in the reviewers’ PDF.

Reviewers' comments:

Reviewer's Responses to Questions

**Comments to the Author**

1. Is the manuscript technically sound, and do the data support the conclusions?

Reviewer #1: Partly

Reviewer #2: Partly

2. Has the statistical analysis been performed appropriately and rigorously? 

Reviewer #1: Yes

Reviewer #2: No

3. Have the authors made all data underlying the findings in their manuscript fully available?

Reviewer #1: Yes

Reviewer #2: Yes

4. Is the manuscript presented in an intelligible fashion and written in standard English?

Reviewer #1: Yes

Reviewer #2: Yes

5. Review Comments to the Author

Reviewer #1: Dear Authors,

Thank you for submitting the data and analysis regarding a strategy to improve the availability and administration of injectable contraceptives in underserved rural regions lacking supplies and healthcare.

The description of the strategy addresses acceptability, feasibility, and potential effectiveness through a training program led by experienced users and an accessibility strategy for self-administration of injectable contraceptives.

Relevant effects were achieved according to the respondents and expected outcomes following the training and improvements in accessibility in situations where the HSA was frequently absent.

Consideration of whether the effectiveness of contraception in a population was evaluated may be beyond the scope of the present evaluation design and should be avoided in the manuscript.

Title:

It is suggested to avoid the concept of effectiveness as there is no adequate population comparator nor an assessment of its contraceptive effect within the present report.

Abstract:

The comments regarding effectiveness also apply here.

Methods:

Regarding Consent, please address the considerations for Medroxyprogesterone Acetate concerning bone mineral density and potential long-term effects on the breast. If it was informed to patients.

It would be appropriate to describe how the temperature of medications was ensured during bicycle transport, as well as the separation from food items.

The direct compensation provided to survey participants by the interviewers could be a variable influencing the acquired responses.

The indirect compensation is linked to the decision to accept the injection. The reasons of individuals who did not accept the medication administration were not explored.

The strategy is evaluated through self-reporting of injectable medication administration instructed by an experienced user but not validated by a healthcare professional at any point, which may have implications for the ultimate objective.

Contraceptive effectiveness in a population cannot be measured with the proposed design, particularly without adequate monitoring of transport temperature and proper review of the administration technique by a healthcare professional.

Discussion:

A strategy that could be sustainable without continuous support is suggested. However, the implementation of the present strategy relies on HSA time (where it was not possible to work with all due to their other activities), the provision of a bicycle in good condition, the provision of lunch allowances, and verification of whether users persist even without compensation for their participation. The aforementioned suggests that the present strategy also requires ongoing investment and sustainability.

Overall Recommendation:

In general, the manuscript with the description of the strategy is suitable for publication with the suggested revisions.

Reviewer #2: Reviewer’s Comments:

Journal: PLOS ONE

Manuscript title: Feasibility, acceptability, and potential effectiveness of a human-centered design-derived intervention to improve community health workers’ contraception outreach in rural Malawi.

First, thank you for the opportunity to review this manuscript. I will say this is a well-structured and timely study addressing a critical gap in contraceptive access in rural Malawi using a human-centered design approach. The mixed-methods design is appropriate. The triangulation of data is also commendable. However, the manuscript requires some improvements in some areas to meet the standard of a peer-reviewed publication. Below are my suggestions for improvement.

Title and Abstract:

• Page 1, Lines 5–6: The title is clear, but slightly long and can confuse. Remember, your work/data reads a pilot study. I will suggest you consider shortening it to read: “Improving Contraception Outreach through Human-Centered Design: A Pilot Study in Rural Malawi.”

• Page 2, Lines 30–54 (Abstract). The abstract is detail but exceeds typical word limits for many journals. It is above 300 words and its current state stresses readers. I will suggest you consider trimming descriptive content, especially in the “Methods” and “Results” subsections, without losing key details. Make it smart and interesting to readers.

Introduction:

• Page 4, Lines 58–87: The introduction somewhat effectively outlines the problem but needs a clearer articulation of the knowledge gap. The authors should more explicitly state why previous approaches have not succeeded and how Ndingathe uniquely addresses this. You may wish to address the question of what’s novel beyond combining peer support and logistics support. Also, what ha previous studies focused on and what gap in the body of knowledge is this study bridging or filling.

• Additionally, please ensure that in-text references follow the PLOS ONE style, with reference numbers inserted before the full stop. This also apply to other sections of the manuscript. I recommend you consulting the PLOS ONE journal's author submission guidelines for detailed instructions on formatting and referencing

Theoretical Framework:

• Page 7, Lines 135–13: The “Contraceptive Agency Framework” is referenced, but there is no adequate explanation of what the framework entails. I will suggest you provide a concise summary of the framework or a figure caption for “Figure 1” that explains the mechanisms and constructs. This will help readers to understand better and be carried along.

Study Design and Methodology:

• Page 9, between lines 153–159: The use of a “reduced intervention package” as a quasi-comparator group is mentioned but not well developed. It doesn’t carry the readers along clearly as expected. Please expand the rationale and design of this comparison arm. If I may ask, were outcomes stratified by intervention exposure level (with vs. without EUs)? If so, please present those comparisons.

• Page 10, Around Lines 168–173: The sampling strategy is not fully described. How were the 450 clients selected? Was there any randomization or consecutive sampling? Please clarify recruitment criteria to assess risk of selection bias. This is very important, please.

• Page 12, Lines 230–236: The observation methodology you used slightly lacks detail. How were the 20 outreach sessions selected? Were observers standardized or trained for consistency? What made up the observation template? Was it only the number of clients that visited by time of day? Please address these in the methodology section.

Data Analysis:

• Page 13, Lines 237–244: On your analysis, while the statistical tests (Wilcoxon, t-tests) seem appropriate and I don’t have problem with you using them, but the manuscript does not address potential confounders or clustering effects (e.g., HSAs nested within districts). Could you please address this? Please address the limitation of not controlling for clustering or interdependence in responses. If clustering wasn't accounted for, acknowledge this as a limitation.

• Please indicate also how missing data were handled (e.g., non-responses, incomplete forms) in your study/analysis.

Results:

• The use of chart in the presentation of your paired t-test may seem pictorial but lacking some key ingredients which are very important, and has pushed me to interrogate the presentation and completeness your statistical test results, especially given the reliance on paired t-tests and Wilcoxon signed-rank tests to establish pre-post differences.

• I notice inadequate presentation of paired t-test and Wilcoxon results across the results. For example, on Page 20, Figure 3 (HSAs’ role conflict, overload, job satisfaction), Page 21, Figure 5 (Client-reported fear before/after interacting with experienced users) and other Figures where you used paired t-tests and Wilcoxon signed-rank tests. The manuscript only presents mean values and p-values via bar charts. While this is helpful for visual readers, bar charts alone are not sufficient to fully interpret statistical results. In fact, key statistical reporting elements are missing, including:

Mean difference (or median difference for Wilcoxon)

Standard deviation or standard error

95% Confidence Interval (CI) around the difference

Test statistics (e.g., t-value, z-score, or W-statistic for Wilcoxon)

Sample size (n) per comparison group clearly labelled

Without this information, readers cannot assess the precision, effect size, or practical significance of the findings, which you will agree with me if you carefully check through your results presentations. As you are aware p-values alone which you used tell us whether a difference is statistically significant but not the magnitude or relevance of that difference.

• I will suggest you use Tables instead of charts format. The Table format allows quick comparison, gives effect size estimates, includes sample size, CI, and test details. In fact, it complies with EQUATOR Network (e.g., CONSORT, STROBE) reporting standards.

• If at all visual presentation is retained, annotate bar charts with exact mean/median values, n, and 95% CI bars is preferred. You should include footnotes stating the test used and p-values. Avoid using bar graphs for ordinal scales (e.g., Likert-based fear levels). Boxplots or stacked bar charts showing distribution would be more appropriate for Wilcoxon tests. Despite this, I will still suggest, you use table format to align with scientific writing. It is not a policy brief. It is an original research manuscript.

• You interpretation also need to improve. Add interpretation of Practical Significance. May it not only statistically significant, but meaningful? The authors should discuss what the results mean in practice to be clinically or programmatically relevant?

• Page 15, Line 284: Statement: “...suggesting that offering afternoon clinics was helpful...” This is an inference not strongly supported by comparative data. I would have said authors should consider including pre-intervention time-use data or more rigorous client preference analysis. However, I do hope, a table presentation of the paired test results showing all the statistical test information should address this.

• Page 16, Lines 295–303: The quote from the 22-year-old client is strong. However, please consider providing more thematic synthesis across quotes to highlight commonalities, not just illustrative anecdotes. Using only a quote from a respondent to address this very theme is disturbing. Since the authors used a software for analysis, I was expecting to see thematic analysis running through quotes. Could you address this, please?

• Page 20, Line 369–371 (Fig. 3): The reductions in role conflict and overload are statistically significant, but the clinical or programmatic relevance of these changes is not discussed. In addressing this, please provide effect sizes or interpretation of whether these are considered “meaningful” changes in workload and motivation. I want to believe your table presentation of results will assist in this aspect also.

Discussion:

• Page 26, Lines 499–507: The discussion reiterates findings but still lacks important reflection on what did not work. For example, the issue of delayed lunch allowances (page 18, lines 346–348) should be discussed more critically in terms of sustainability and implementation fidelity.

• On Page 27, Lines 525–529, the authors mention continued mnemonic use after the pilot, which is encouraging. However, it's unclear whether this was tracked systematically or anecdotal. Please clarify whether this finding was emergent from interviews or formally evaluated.

• Please also support your findings with previous studies showing aspects of agreements and disagreements with past studies. This is very important.

Limitations:

• Page 28, Lines 559–568: The limitations are appropriately acknowledged which are good, but the absence of a control group should be emphasized more strongly as a major constraint to attributing effects to the intervention. I know that you mentioned control group but clearly state that is a major constraint to attributing effects to the intervention.

Writing and Clarity:

Here, I will say that there are some aspects of the manuscript that contain grammatical errors or awkward phrasing, for example:

• Page 9, Line 108: “so that there would not be any overlap”. Consider rephrasing to read “to avoid overlap in outreach areas.”

• Page 27, Line 546: “...generally well-received”. Please also consider saying “broadly acceptable and culturally appropriate.”

I will suggest that a thorough copyedit is needed to correct minor grammar and enhance readability.

Figures and Tables:

• For your Figure 1, I will suggest you ensure the figure is well-labelled and visually communicates the causal logic between intervention components and contraceptive agency outcomes.

• For your Tables 1 and 2 (Page 14), though they are information, I will still suggest you consider including a footnote indicating how missing data were handled (e.g., non-responses, incomplete forms). I have also included this also be addressed in methodology.

References:

Again, please ensure that in-text references follow the PLOS ONE style, with reference numbers inserted before the full stop. Additionally, in the reference list, include the full DOI (https://doi.org/) or web address where applicable. I recommend consulting the journal's author submission guidelines for detailed instructions on formatting and referencing

6. PLOS authors have the option to publish the peer review history of their article (what does this mean?). If published, this will include your full peer review and any attached files.

Reviewer #1: **Yes: **MD PhD Daniel Humberto Mendez Lozano

Reviewer #2: **Yes: **Turnwait Otu Michael

---

## [Author Response · Author response to Decision Letter 1]

28 Jul 2025

Response to reviewers: Improving Contraception Outreach through Human-Centered Design: A Pilot Study of the Ndingathe (“I Can”) intervention in Rural Malawi

Date of resubmit: July 10, 2025

Responses to Academic Editor Comments

1. Abstract: Add a sentence or two about data analysis.

Response: We added a sentence about data analysis on page 2 line 40-41 of the abstract.

New text: “We analyzed quantitative data using descriptive and inferential statistics. We conducted a thematic analysis of qualitative data.”

2. Introduction, Page 4, Lines 67-69: The current text states, "These workers are often people’s first point of contact for health-related issues and are instrumental in promoting and providing contraceptive services, among many other primary care services." This should be reworded to clarify that Health Surveillance Assistants (HSAs) are the first point of contact for health-related issues in areas far from health centres (Primary health care delivery settings). This distinction is important because medical assistants are typically the first point of contact for health-related problems in rural areas. The primary role of HSAs is to promote health and prevent illnesses.

Response: Thank you for this important consideration. We reworded to clarify that HSAs are the first point of contact for people who living in rural areas on page 4 Line 70 of the introduction.

New text in italics: “In rural areas far from primary care clinics, HSAs are often people’s first point of contact for health-related issues and are instrumental in providing contraceptive services.”

3. Methods (Description of the intervention)- correct the typo on page 7, line 136, where 'Ndingathe' has been written as 'Ndingathe'. Also, could you please provide more information about the contents of the observation template? Was the template structured or unstructured?

Response: We corrected the typo on page 7 line 141. We also added one sentence to page 11 lines 247-248 to provide more information about the contents of the structured observation template.

New text: “A structured observation template guided data collection of the number of clients arriving in the morning versus the afternoon.”

4. Data Analysis: Please provide some details on how the framework analysis was used to analyse the transcripts and develop the themes. Could you also explain how data from observation were analysed?

Response: Thank you for raising this important point. We provided details on how the framework analysis was used to analyze the transcripts and develop themes on page 12 lines 270-276.

New text: “After transcription, two researchers (JV and KG) independently reviewed the interview transcripts to familiarize themselves with the data. Relevant excerpts were extracted and organized into a matrix around the pilot intervention's key components: the work planning tool, bicycles and lunch allowances, the mnemonic “Sakufima,” and engagement with EUs. Within each element, excerpts were categorized according to feasibility and acceptability. Five researchers (IM, TJ, MN, JV, MK) were each assigned one component and used the matrix to develop themes. Organizing the data this way enabled systematic comparison across participant groups (e.g., clients, HSAs, stakeholders).”

Response: We also added a sentence to describe how the observation data was analyzed on page 11 lines 257-258.

New text: “For observational data, we calculated descriptive statistics of the number of women observed attending outreach clinics in the morning versus the afternoon.”

Responses Reviewer #1 Comments

Dear Authors,

Thank you for submitting the data and analysis regarding a strategy to improve the availability and administration of injectable contraceptives in underserved rural regions lacking supplies and healthcare. The description of the strategy addresses acceptability, feasibility, and potential effectiveness through a training program led by experienced users and an accessibility strategy for self-administration of injectable contraceptives. Relevant effects were achieved according to the respondents and expected outcomes following the training and improvements in accessibility in situations where the HSA was frequently absent. Consideration of whether the effectiveness of contraception in a population was evaluated may be beyond the scope of the present evaluation design and should be avoided in the manuscript.

Response: Thank you for providing these detailed and thoughtful comments on our manuscript and allowing us the opportunity to clarify our study design. We have provided our responses below, including the text from our updated manuscript to highlight what we changed based on your feedback. We sought to study the preliminary effectiveness of the intervention, Ndingathe or ICAN, not the contraceptive method, DMPA-SC. We have clarified this in the following sections:

New text: In the introduction on page 5 lines 103-106 “By intentionally combining system-level supports for HSAs with peer support for women interested in self-injection, Ndingathe offers a novel, holistic approach to improving contraceptive access and strengthening women’s ability to choose self-injection if they prefer this method.”

New text in italics: In the methods section on page 7 lines 149-154 “The Ndingathe theory of action (Figure 1) depicts the two main pathways by which we posit that the intervention will improve women’s contraceptive agency, defined as agency related to making and acting on contraceptive decisions [25] and increasing use of self-injection [22]. First, optimizing the HSA workflow is intended to improve service accessibility. Second, strengthening self-injection support is intended to increase clients’ self-efficacy and perceived control over contraceptive decision-making and self-efficacy to self-inject.”

5. Title: It is suggested to avoid the concept of effectiveness as there is no adequate population comparator nor an assessment of its contraceptive effect within the present report.

Response: We adjusted the title to avoid the concept of effectiveness on page 1 lines 5-6.

New text: The title was changed to Improving Contraception Outreach through Human-Centered Design: A Pilot Study of the Ndingathe (“I Can”) intervention in Rural Malawi per Reviewer #2’s recommendation.

6. Abstract: The comments regarding effectiveness also apply here.

Response: We clarified that we aimed to study the preliminary effectiveness of the Ndingathe intervention, so we retained language about this in the abstract on page 2, line 37. “To assess the feasibility, acceptability, and potential effectiveness of Ndingathe, we collected…”

Methods:

7. Regarding Consent, please address the considerations for Medroxyprogesterone Acetate concerning bone mineral density and potential long-term effects on the breast. If it was informed to patients.

Response: Clinical information about contraception was beyond the scope of our consent form, which focused on informing women of procedures and potential risks involved in taking part in a survey or interview with the research team. Our study focused on understanding women’s experiences with the Ndingathe intervention that all contraception clients (not just those who participated in our study) were receiving during the time period in which it was being piloted in the study districts.

8. It would be appropriate to describe how the temperature of medications was ensured during bicycle transport, as well as the separation from food items.

Response: Thank you for raising this point. Information on medication storage is beyond the scope of our intervention and pilot study, which did not alter HSAs’ transport of medications. Further, HSAs did not typically transport food; the allowance provided to them as part of the intervention was used by them to purchase food at a local restaurant near the outreach clinic. See text page 7 134-135 lines: “The lunch allowance given to HSAs as part of the intervention was intended to be used to purchase food at a local restaurant near the outreach clinic.”

9. The direct compensation provided to survey participants by the interviewers could be a variable influencing the acquired responses.

Response: We appreciate this concern. We included a sentence to account for courtesy bias from incentives given to survey participants in the Limitations section on page 27, line 588-590.

New text: “Lastly, we reimbursed clients, EUs, and stakeholders for their time answering interview questions, which may have influenced their response and introduced courtesy bias.”

10. The indirect compensation is linked to the decision to accept the injection. The reasons of individuals who did not accept the medication administration were not explored.

Response: Choice of injection was not an eligibility criterion for joining the study. Women who had interacted with HSAs or experienced users during an outreach clinic were eligible for surveys and interviews regardless of their contraceptive choice. We did not collect information on contraceptive choice as that was beyond the scope of our research questions.

11. The strategy is evaluated through self-reporting of injectable medication administration instructed by an experienced user but not validated by a healthcare professional at any point, which may have implications for the ultimate objective.

Response: We revised the sentence on page 7, line 141-143 to describe that healthcare professionals were also present during training.

New text: “In the pilot, the EU’s role complemented the training provided by HSAs, who ultimately retained authority over validating women’s correct SI DMPA-SC technique during training.”

12. Contraceptive effectiveness in a population cannot be measured with the proposed design, particularly without adequate monitoring of transport temperature and proper review of the administration technique by a healthcare professional.

Response: We did not study contraceptive effectiveness and do not report on it in this manuscript. We evaluated the potential promise of the Ndingathe intervention and share results on the potential effectiveness of the intervention to improve contraceptive access and strengthen women’s ability to choose self-injection if it is a method aligned with their preferences.

13. Discussion: A strategy that could be sustainable without continuous support is suggested. However, the implementation of the present strategy relies on HSA time (where it was not possible to work with all due to their other activities), the provision of a bicycle in good condition, the provision of lunch allowances, and verification of whether users persist even without compensation for their participation. The aforementioned suggests that the present strategy also requires ongoing investment and sustainability.

Response: Thank you for highlighting this important point. To better reflect the fact that scale-up of this intervention would require more information as to the resources required for sustainability, we have added a line in the Discussion: page 27 Lines 598-601

New text in italics: “While these initial results are encouraging, further testing is warranted to establish the solution's effectiveness in achieving its intended outcomes, such as improved contraceptive agency, and to evaluate the minimum amount of resources required to ensure sustainability of the model.”

Overall Recommendation:

In general, the manuscript with the description of the strategy is suitable for publication with the suggested revisions.

Responses to Reviewer #2’s Comments

First, thank you for the opportunity to review this manuscript. I will say this is a well-structured and timely study addressing a critical gap in contraceptive access in rural Malawi using a human-centered design approach. The mixed-methods design is appropriate. The triangulation of data is also commendable. However, the manuscript requires some improvements in some areas to meet the standard of a peer-reviewed publication. Below are my suggestions for improvement.

Response: Thank you for your informative and thoughtful review and instructive comments that allowed us to strengthen our manuscript. We have provided our responses below, including the text from our updated manuscript to highlight what we changed based on your feedback.

Title and Abstract:

14. Page 1, Lines 5–6: The title is clear, but slightly long and can confuse. Remember, your work/data reads a pilot study. I will suggest you consider shortening it to read: “Improving Contraception Outreach through Human-Centered Design: A Pilot Study in Rural Malawi.”

Response: The title was changed to reflect a pilot study and include the name of the intervention. The title was changed to Improving Contraception Outreach through Human-Centered Design: A Pilot Study of the Ndingathe (“I Can”) intervention in Rural Malawi.

15. Page 2, Lines 30–54 (Abstract). The abstract is detail but exceeds typical word limits for many journals. It is above 300 words and its current state stresses readers. I will suggest you consider trimming descriptive content, especially in the “Methods” and “Results” subsections, without losing key details. Make it smart and interesting to readers.

Response: We edited the abstract and tightened the language to make it more clear and engaging for readers.

Introduction:

16. Page 4, Lines 58–87: The introduction somewhat effectively outlines the problem but needs a clearer articulation of the knowledge gap. The authors should more explicitly state why previous approaches have not succeeded and how Ndingathe uniquely addresses this. You may wish to address the question of what’s novel beyond combining peer support and logistics support. Also, what have previous studies focused on and what gap in the body of knowledge is this study bridging or filling.

Response: We have revised the introduction to more clearly articulate the knowledge gap our study addresses. Specifically, we now include a review of prior interventions to highlight how most existing efforts have focused on either improving awareness raising of the product or provider training—often neglecting the systemic challenges HSAs encounter in conducting outreach and the psychosocial support women need to overcome well-documented fear of self-injection.

New text: “Despite the potential of DMPA-SC to ease contraceptive access barriers, the percentage of DMPA-SC users who self-inject in Malawi remains low (21% as of 2021), and qualitative research has identified fear of SI as a major concern among women [12,17]. Previous contraceptive counseling and message-based counseling interventions have shown promise for spreading awareness of the option to self-inject DMPA-SC, but the positive impact of these interventions on improving the quality of provider-patient interaction during contraceptive counseling is mixed [18,19]. Approaches to simultaneously address service delivery challenges faced by HSAs serving rural areas and the psychosocial needs of clients to follow through on their interest in self-injection is lacking. Other research has highlighted the need to not only ensure the availability of this contraceptive method option, but also to ensure women have support to overcome fears of injecting themselves and support for users to deal with potential side effect concerns, desire to switch/stop methods or other concerns [20,21].”

17. Additionally, please ensure that in-text references follow the PLOS ONE style, with reference numbers inserted before the full stop. This also apply to other sections of the manuscript. I recommend you consulting the PLOS ONE journal's author submission guidelines for detailed instructions on formatting and referencing

Response: We revised in-text references per the PLOS ONE journal’s author submission guidelines.

18. Theoretical Framework: Page 7, Lines 135–13: The “Contraceptive Agency Framework” is referenced, but there is no adequate explanation of what the framework entails. I will suggest you provide a concise summary of the framework or a figure caption for “Figure 1” that explains the mechanisms and constructs. This will help readers to under

---

## [Decision Letter · Decision Letter 1]

27 Aug 2025

Improving Contraception Outreach through Human-Centered Design: A Pilot Study of the Ndingathe (“I Can”) intervention in Rural Malawi

PONE-D-25-06248R1

Dear Dr. Vallin,

We’re pleased to inform you that your manuscript has been judged scientifically suitable for publication and will be formally accepted for publication once it meets all outstanding technical requirements.

Kind regards,

Fatch Welcome Kalembo, Ph.D

Academic Editor

PLOS ONE

Additional Editor Comments (optional):

Reviewers' comments:

Reviewer's Responses to Questions

**Comments to the Author**

1. If the authors have adequately addressed your comments raised in a previous round of review and you feel that this manuscript is now acceptable for publication, you may indicate that here to bypass the “Comments to the Author” section, enter your conflict of interest statement in the “Confidential to Editor” section, and submit your "Accept" recommendation.

Reviewer #1: All comments have been addressed

2. Is the manuscript technically sound, and do the data support the conclusions?

Reviewer #1: Yes

3. Has the statistical analysis been performed appropriately and rigorously? 

Reviewer #1: Yes

4. Have the authors made all data underlying the findings in their manuscript fully available?

Reviewer #1: Yes

5. Is the manuscript presented in an intelligible fashion and written in standard English?

Reviewer #1: Yes

6. Review Comments to the Author

Reviewer #1: Dear Authors,

I have reviewed your revised manuscript and am pleased with the thoroughness of your response. The changes you have made have significantly improved the data analysis for the intervention under study.

Congratulations on the progress. I look forward to seeing your work published.

7. PLOS authors have the option to publish the peer review history of their article (what does this mean?). If published, this will include your full peer review and any attached files.

Reviewer #1: **Yes: **MD PhD Daniel Humberto Mendez Lozano

---

## [Editor Report · Acceptance letter]

PONE-D-25-06248R1

PLOS ONE

Dear Dr. Vallin,

I'm pleased to inform you that your manuscript has been deemed suitable for publication in PLOS ONE. Congratulations! Your manuscript is now being handed over to our production team.

Kind regards,

on behalf of

Dr. Fatch Welcome Kalembo

Academic Editor

PLOS ONE